# Heterologous Production of Isopropanol Using Metabolically Engineered *Acetobacterium woodii* Strains

**DOI:** 10.3390/bioengineering10121381

**Published:** 2023-11-30

**Authors:** Franziska Höfele, Teresa Schoch, Catarina Oberlies, Peter Dürre

**Affiliations:** 1Institute of Molecular Biology and Biotechnology of Prokaryotes, Ulm University, 89081 Ulm, Germany; 2Institute of Microbiology and Biotechnology, Ulm University, 89081 Ulm, Germanycatarina.oberlies@gmail.com (C.O.); peter.duerre@uni-ulm.de (P.D.)

**Keywords:** acetogens, *Acetobacterium woodii*, anaerobic fermentation, Wood–Ljungdahl pathway, solvents, isopropanol, metabolic engineering

## Abstract

The depletion of fossil fuel resources and the CO_2_ emissions coupled with petroleum-based industrial processes present a relevant issue for the whole of society. An alternative to the fossil-based production of chemicals is microbial fermentation using acetogens. Acetogenic bacteria are able to metabolize CO or CO_2_ (+H_2_) via the Wood–Ljungdahl pathway. As isopropanol is widely used in a variety of industrial branches, it is advantageous to find a fossil-independent production process. In this study, *Acetobacterium woodii* was employed to produce isopropanol via plasmid-based expression of the enzymes thiolase A, CoA-transferase, acetoacetate decarboxylase and secondary alcohol dehydrogenase. An examination of the enzymes originating from different organisms led to a maximum isopropanol production of 5.64 ± 1.08 mM using CO_2_ + H_2_ as the carbon and energy source. To this end, the genes *thlA* (encoding thiolase A) and *ctfA/ctfB* (encoding CoA-transferase) of *Clostridium scatologenes*, *adc* (encoding acetoacetate decarboxylase) originating from *C. acetobutylicum* and *sadH* (encoding secondary alcohol dehydrogenase) of *C. beijerinckii* DSM 6423 were employed. Since bottlenecks in the isopropanol production pathway are known, optimization of the strain was investigated, resulting in a 2.5-fold increase in isopropanol concentration.

## 1. Introduction

Isopropanol, which is also known as isopropyl alcohol, is an important solvent and chemical intermediate for a variety of industrial branches. It is widely used as a solvent, cleaner or as a raw material for different derivatives, and it has gained interest since the coronavirus pandemic (COVID-19) due to its biocidal properties [1,2]. As isopropanol is typically produced via the hydration of propylene, and propylene is synthesized via the steam cracking of petroleum feedstocks, the production process is fossil-based [1,3]. Fossil fuel resources are shrinking, and petroleum-based processes are coupled with the excessive release of greenhouse gas emissions; therefore, alternative production processes need to be established [4]. A promising alternative is found in fermentation using acetogenic bacteria. Acetogens, which are widespread throughout a huge variety of genera, employ the Wood–Ljungdahl pathway (WLP) for the fixation of CO or CO_2_ (together with H_2_) by producing acetate as the main product [5,6]. Some acetogens are already used for the industrial production of solvents or alcohols, e.g., LanzaTech is using the organism *Clostridium autoethanogenum* for the commercial production of ethanol in anaerobic fermentation plants [7,8]. The manufacture of high-value chemicals using acetogens has been favored during the past decades, as acetogens utilize greenhouse gas emissions instead of fossil fuels or renewable resources. Thereby, the food-vs.-fuel debate can be avoided. Furthermore, a carbon-neutral way of production can be established [2,9]. The bacterial species *Acetobacterium woodii* comprises the genes of the WLP and produces acetate as the sole product; hence, it is used as a model organism for understanding the WLP as well as for the metabolic engineering of acetogenic bacteria [10,11]. The WLP consists of a methyl and a carbonyl branch. In the carbonyl branch, CO_2_ is taken up via carbon monoxide dehydrogenase (CODH) and converted to CO by oxidation of a reduced ferredoxin (Fd^2−^). CO_2_ is also taken up by the CODH in the methyl branch, followed by the conversion into formate via formate dehydrogenase. In an ATP-dependent step, formate is bound to tetrahydrofolate (THF) by formyl-THF synthetase. Employing the methenyl-THF cyclohydrolase, methylene-THF dehydrogenase and methenyl-THF reductase, formyl-THF is further processed to methyl-THF. After methyl transferase exchanges THF against an iron–sulfur cluster containing corrinoid protein (CoFeS-P), methyl-CoFeS-P is converted together with CO (from the carbonyl branch) to acetyl-CoA by acetyl-CoA synthase [12]. In *A. woodii*, acetyl-coenzmye A (CoA) is converted to the sole product acetate along with the formation of ATP [10].

Over the past decades, *A. woodii* has been metabolically engineered, e.g., for the production of the bioplastic poly-3-hydroxybutyrate, lactate or the chemical intermediate acetone [13,14,15,16]. The heterologous production of acetone includes the plasmid-based expression of the genes *thlA*, encoding a thiolase A, *ctfA/ctfB*, encoding a CoA-transferase, and *adc*, encoding an acetoacetate decarboxylase. Acetyl-CoA is produced via the WLP as a central intermediate and represents the starting point for acetone production. Two moieties of acetyl-CoA are converted to acetoacetyl-CoA via thiolase A. CoA-transferase transfers the CoA to one moiety of acetate by releasing acetoacetate. By cleaving off CO_2_ via the acetoacetate decarboxylase, acetone is produced [15]. During fermentation of the different recombinant *A. woodii* strains for acetone production, small amounts of isopropanol were also monitored. As it is known that the conversion of acetone to isopropanol is feasible via a secondary alcohol dehydrogenase (Sadh), the genome of *A. woodii* was screened for an appropriate enzyme for isopropanol production. Until now, no *sadH* gene was found in the genome of *A. woodii* [16].

In this study, the acetogenic bacterium *A. woodii* was employed for the autotrophic, recombinant production of isopropanol using CO_2_ and H_2_ as the carbon and energy resource. Therefore, the *sadH* genes of *Clostridium beijerinckii* DSM 6423 and *C. ljungdahlii* were chosen as the candidate enzymes for isopropanol production [17,18,19]. Via BLAST analysis, another *sadH* gene was found in the genome of *C. beijerinckii* DSM 15410, which was also tested for recombinant isopropanol production. The production of isopropanol using different anaerobic *Clostridium* species already revealed bottlenecks of acetone and isopropanol production [2,20,21]. Limitations regarding the conversion of acetone to isopropanol can be avoided by the insertion of additional copies of the *sadH* gene or expression of a gene named *hydG*, which can be found in the same operon as *sadH*. The gene *hydG* is annotated in both the *C. beijerinckii* and *C. ljungdahlii* genomes and it is predicted to function as an electron donor protein [20]. Another bottleneck in the acetone production is the efficiency of the CoA-transferase, which can be bypassed by inserting an additional copy of the *ctfA*/*ctfB* gene [2,20].

## 2. Materials and Methods

### 2.1. Bacterial Strains and Media

The bacterial strains used in this study are listed in Table 1. *Escherichia coli* XL1-Blue MRF’ was used for plasmid construction and amplification. Strains of *E. coli* XL1-Blue MRF’ were cultivated in Lysogeny Broth (LB) medium (containing per liter: tryptone 10 g, NaCl 10 g, yeast extract 5 g) with the respective antibiotic aerobically at 37 °C with shaking at 180 rpm [22].

The cultivation of the *A. woodii* DSM 1030 strains was carried out anaerobically at 30 °C in modified DSM medium 135 (containing per liter: NH_4_Cl 0.2 g, KH_2_PO_4_ 1.76 g, K_2_HPO_4_ 8.44 g, yeast extract 3 g, NaHCO_3_ 10 g, HCl-cysteine monohydrate 0.3 g, Na_2_S nonahydrate 0.3 g, trace element solution 2 mL, vitamin solution 2 mL, resazurin 1 mg; trace element solution (per liter): nitrilotriacetate 12.8 g, NaOH 0.5 g, MnCl_2_ tetrahydrate 0.1 g, NaCl 5 g, FeCl_2_ tetrahydrate 2 g, CoCl_2_ hexahydrate 0.2 g, ZnCl_2_ 70 mg, CuCl_2_ dihydrate 2 mg, H_3_BO_3_ 6 mg, Na_2_MoO_4_ dihydrate 36 mg, NiCl_2_ hexahydrate 24 mg, Na_2_SeO_3_ pentahydrate 3 mg, Na_2_WO_4_ dihydrate 4 mg; vitamin solution (per liter): HCl-pyridoxine 50 mg, HCl-thiamine monohydrate 50 mg, riboflavin 50 mg, D-Ca-pantothenate 50 mg, lipoic acid 25 mg, p-aminobenzoate 50 mg, nicotinic acid 50 mg, vitamin B_12_ 25 mg, biotin 25 mg, folic acid 25 mg). The respective antibiotic, 40 mM fructose and 1.3 mM MgSO_4_ were added after autoclaving at 121 °C and 1.2 bar for 15 min [15].

For the cultivation of *C. kluyveri* DSM 555, the DSM 52 medium (containing per liter: K-acetate 10 g, K_2_HPO_4_ 0.31 g, KH_2_PO_4_ 0.23 g, NH_4_Cl 0.25 g, MgSO_4_ heptahydrate 0.2 g, yeast extract 1 g, trace element solution 1 mL, selenite-tungstate solution 1 mL, resazurin 0.5 mg, ethanol 20 mL, Na_2_CO_3_ 1 g, seven vitamins solution 1 mL, HCl-cysteine monohydrate 0.25 g, Na_2_S nonahydrate 0.25 g; trace element solution (per liter): HCl (25%) 10 mL, FeCl_2_ tetrahydrate 1.5 g, ZnCl_2_ 70 mg, MnCl_2_ tetrahydrate 100 mg, H_3_BO_3_ 6 mg, CoCl_2_ hexahydrate 190 mg, CuCl_2_ dihydrate 2 mg, NiCl_2_ hexahydrate 24 mg, Na_2_MoO_4_ dihydrate; selenite-tungstate solution (per liter): NaOH 0.5 g, Na_2_SeO_3_ pentahydrate 3 mg, Na_2_WO_4_ dihydrate 4 mg; seven vitamins solution (per liter): vitamin B_12_ 100 mg, p-aminobenzoate 80 mg, biotin 20 mg, nicotinic acid 200 mg, D-Ca-pantothenate 100 mg, HCl-pyridoxine 300 mg, HCl-thiamine dihydrate 200 mg) was used anaerobically at 37 °C. Before autoclaving at 121 °C and 1.2 bar for 15 min, the pH was adjusted to 7.

*C. scatologenes* DSM 757 and *C. ljungdahlii* DSM 13528 were cultivated anaerobically at 37 °C in modified Tanner medium (containing per liter: MES 10 g, yeast extract 2 g, HCl-cysteine monohydrate 0.5 g, mineral solution 10 mL, trace element solution 10 mL, vitamin solution 10 mL, resazurin 1 mg; mineral solution (per liter): NH_4_Cl 100 g, NaCl 80 g, KCl 10 g, KH_2_PO_4_ 10 g, MgSO_4_ heptahydrate 20 g, CaCl_2_ dihydrate 4 g; trace element solution (per liter): nitrilotriacetate 2 g, MnSO_4_ monohydrate 1 g, Fe(NH_4_)_2_(SO_4_)_2_ hexahydrate 0.8 g, CoCl_2_ hexahydrate 0.2 g, ZnSO_4_ heptahydrate 0.2 g, NiCl_2_ hexahydrate 20 mg, CuCl_2_ dihydrate 20 mg, Na_2_MoO_4_ dihydrate 20 mg, Na_2_SeO_3_ pentahydrate 20 mg, Na_2_WO_4_ dihydrate 20 mg; vitamin solution (per liter): HCl-pyridoxine 10 mg, HCl-thiamine monohydrate 5 mg, riboflavin 5 mg, D-Ca-pantothenate 5 mg, lipoic acid 5 mg, p-aminobenzoate 5 mg, nicotinic acid 5 mg, vitamin B_12_ 5 mg, biotin 2 mg, folic acid 2 mg) [23]. After the pH was set to 6, the medium was autoclaved at 121 °C and 1.2 bar for 15 min. After autoclaving, 40 mM of fructose were added as the carbon source.

The cultivation of *C. beijerinckii* DSM 6423 was carried out anaerobically at 37 °C in yeast tryptone glucose (YTG) medium (containing per liter: NaCl 5 g, yeast extract 10 g, tryptone 16 g, glucose 20 g, resazurin 1 mg).

*C. beijerinckii* DSM 15410 was cultivated anaerobically at 37 °C in DSM 104b medium (containing per liter: trypticase peptone 5 g, meat peptone (pepsin-digested) 5 g, yeast extract 10 g, salt solution 40 mL, resazurin 1 mg, HCl-cysteine monohydrate 0.5 g, glucose 5 g; salt solution (per liter): CaCl_2_ dihydrate 0.25 g, MgSO_4_ heptahydrate 0.5 g, K_2_HPO_4_ 1 g, KH_2_PO_4_ 1 g, NaHCO_3_ 10 g, NaCl 2 g).

### 2.2. Construction of Plasmids

Standard molecular cloning techniques were carried out according to established protocols [22]. For the construction of the acetone production plasmid pJIR750_ac2t2, the vector pJIR750_ac2t1 was digested with the restriction enzymes *Sal*I and *Bam*HI [16]. The thiolase gene (*thlA*, Locus Tag: CKL_3698) of *C. kluyveri* and the promoter P*_thlA_* originating from *C. acetobutylicum* were ligated into the digested vector resulting in the plasmid pJIR750_ac2t2. For the construction of the plasmid pJIR750_ac3t3, pJIR750_ac2t2 was digested using the restriction enzymes *Sal*I and *Kpn*I. The acetoacetyl-CoA:acetate/butyrate CoA transferase genes (*ctfA/ctfB*, Locus tag: EG59DRAFT_00772-00773) and the thiolase A gene (*thlA*, Locus tag: EG59DRAFT_00774) of *C. scatologenes* were inserted into the vector.

For isopropanol production, the plasmids pJIR750_ac1t1, pJIR750_ac1t2, pJIR750_ac2t1, pJIR750_ac2t2 and pJIR750_ac3t3 were digested with the restriction enzyme *Eco*RI. The secondary alcohol dehydrogenase genes (*sadH*, Locus Tags: CLOBI_40010, K684DRAFT_02545 and CLJU_c24860, respectively) of *C. beijerinckii* DSM 6423, *C. beijerinckii* DSM 15410 and *C. ljungdahlii* were inserted into the different vectors, resulting in 15 different isopropanol production plasmids listed in Table 2.

For the construction of the two-plasmid system, the vector pMTL83251 was digested with the restriction enzymes *Bam*HI and *Kpn*I [24]. The gene *hydG* (Locus Tag: CLOBI_40020) and the *sadH-hydG* gene cluster (Locus tag: CLOBI_40010–40020) with an intergenic region were inserted under control of the P*_thlA_* promoter resulting in the plasmids pMTL83251_PthlA_h1 and pMTL83251_PthlA_sh1, respectively. In a second step, pMTL83251_PthlA_sh1 was digested with the restriction enzymes *Nco*I and *Aat*II, and the genes *ctfA/ctfB* (Locus tag: EG59DRAFT_00772–00773) of *C. scatologenes* were inserted, resulting in the plasmid pMTL83251_PthlA_sh1c3.

**Table 2 bioengineering-10-01381-t002:** Plasmids used in this work.

Plasmid	Characteristics	Origin
pJIR750	Cm^r^, pMB1 ori^−^, *lacZ*, pIP404 ori^+^	Bannam and Rood (1993) [25]
pJIR750_ac1t1	pJIR750, P*_thlA_*, *adc*, *ctfA/ctfB* and *thlA* of *C. acetobutylicum*	Arslan et al. (2022) [15]
pJIR750_ac1t2	pJIR750, P*_thlA_*, *adc* and *ctfA/ctfB* of *C. acetobutylicum*, *thlA* of *C. kluyveri*	Arslan et al. (2022) [15]
pJIR750_ac2t1	pJIR750, P*_thlA_*, *adc* and *thlA* of *C. acetobutylicum*, *ctfA/B* of *C. aceticum*	Arslan et al. (2022) [15]
pJIR750_ac2t2	pJIR750, P*_thlA_*, *adc* of *C. acetobutylicum*, *ctfA/ctfB* of *C. aceticum*, *thlA* of *C. kluyveri*	This work
pJIR750_ac3t3	pJIR750, P*_thlA_*, *adc* of *C. acetobutylicum*, *ctfA/ctfB* and *thlA* of *C. scatologenes*	This work
pJIR750_ac4t4	pJIR750, P*_thlA_*, *adc* of *C. acetobutylicum*, *ctfA/ctfB* and *thlA* of *C. scatologenes*	This work
pJIR750_ac1t1s1	pJIR750_ac1t1, *sadH* of *C. beijerinckii* DSM 6423	This work
pJIR750_ac1t1s2	pJIR750_ac1t1, *sadH* of *C. beijerinckii* DSM 15410	This work
pJIR750_ac1t1s3	pJIR750_ac1t1, *sadH* of *C. ljungdahlii*	This work
pJIR750_ac1t2s1	pJIR750_ac1t2, *sadH* of *C. beijerinckii* DSM 6423	This work
pJIR750_ac1t2s2	pJIR750_ac1t2, *sadH* of *C. beijerinckii* DSM 15410	This work
pJIR750_ac1t2s3	pJIR750_ac1t2, *sadH* of *C. ljungdahlii*	This work
pJIR750_ac2t1s1	pJIR750_ac2t1, *sadH* of *C. beijerinckii* DSM 6423	This work
pJIR750_ac2t1s2	pJIR750_ac2t1, *sadH* of *C. beijerinckii* DSM 15410	This work
pJIR750_ac2t1s3	pJIR750_ac2t1, *sadH* of *C. ljungdahlii*	This work
pJIR750_ac2t2s1	pJRI750_ac2t2, *sadH* of *C. beijerinckii* DSM 6423	This work
pJIR750_ac2t2s2	pJIR750_ac2t2, *sadH* of *C. beijerinckii* DSM 15410	This work
pJIR750_ac2t2s3	pJIR750_ac2t2, *sadH* of *C. ljungdahlii*	This work
pJIR750_ac3t3s1	pJIR750_ac3t3, *sadH* of *C. beijerinckii* DSM 6423	This work
pJIR750_ac3t3s2	pJIR750_ac3t3, *sadH* of *C. beijerinckii* DSM 15410	This work
pJIR750_ac3t3s3	pJIR750_ac3t3, *sadH* of *C. ljungdahlii*	This work
pMTL83251	Em^r^, ColE1 ori^−^, *lacZ*, pCB102 ori^+^, *traJ*	Heap et al. (2009) [24]
pMTL83251_PthlA_h1	pMTL83251, P*_thlA_*, *hydG* of *C. beijerinckii* DSM 6423	This work
pMTL83251_PthlA_sh1	pMTL83251, P*_thlA_*, *sadH-hydG* gene cluster with intergenic region of *C. beijerinckii* DSM 6423	This work
pMTL83251_PthlA_sh1c3	pMTL83251_PthlA_sh1, *ctfA/ctfB* of *C. scatologenes*	This work

### 2.3. Isolation of Genomic and Plasmid DNA

The genomic DNA of Gram-positive organisms was isolated using the MasterPure^TM^ Gram-Positive DNA Purification Kit (Epicentre, Madison, WI, USA) according to the manufacturer’s instructions. The isolation of plasmid DNA from E. coli strains was performed using the Zyppy^TM^ Plasmid Miniprep Kit (ZYMO Research Europe GmbH, Freiburg, Germany) according to the manufacturer’s instructions.

### 2.4. Transformation of A. woodii

For the preparation of electrocompetent *A. woodii* cells and transformation, the recently published protocol of Hoffmeister et al. was used [15]. Transformation was verified by the isolation of genomic DNA and 16S rDNA amplification via PCR. Subsequently, the amplified DNA fragment was sent for sequencing. Using the retransformation of *E. coli* XL1-Blue MRF’, plasmid integrity was verified.

### 2.5. Growth Conditions of Batch Experiments

Growth experiments with *A. woodii* were performed anaerobically at 30 °C in a modified DSM 135 medium without and with shaking at 130 rpm for heterotrophic and autotrophic experiments, respectively. For heterotrophic cultivation, 50 mL of medium were filled into 125 mL serum bottles (SGD Pharma, Paris, France), and 60 mM fructose were added as the carbon and energy source. Autotrophic growth was carried out in 50 mL of medium filled into 500 mL serum bottles (SGD Pharma, Paris, France), and the gas phase was exchanged to 1 bar CO_2_ + H_2_ (33% + 67%, MTI Industriegase AG, Elchingen, Germany). At a minimal pressure of 0.3 bar, the gas phase was replenished during growth experiments.

For adaption to the respective growth conditions, the *A. woodii* strains were transferred once into fresh medium prior to the growth experiments. Experiments were performed as biological triplicates. Integrity of the plasmid and strain verification was carried out before and after the growth experiments as described above.

OD_600_, pH, substrate consumption and end-product concentrations were measured during growth.

### 2.6. Analytics

#### 2.6.1. Optical Density and pH Measurements

The growth of *A. woodii* was monitored by means of the optical density (OD_600_: absorption of the culture at 600 nm) and pH, measured as recently described [13].

#### 2.6.2. High-Performance Liquid Chromatography

Fructose consumption and acetate production during the heterotrophic and autotrophic growth of recombinant *A. woodii* strains were measured via HPLC analysis. Therefore, 500 µL of cell suspension was centrifuged (30 min at 4 °C and 13,000× *g*), the supernatant was transferred to a glass vial (CS-Chromatographie Service GmbH, Langerwehe, Germany) and sealed with a crimp cap (BGB Analytik Vertrieb, Rheinfelden, Germany). For the detection of fructose and acetate, the Agilent 1260 Infinity Series HPLC System (Agilent Technologies, Santa Clara, CA, USA) equipped with a diode array detector, a refractive index detector and the CS-Chromatographie organic acid column with a length of 150 mm (CS-Chromatographie Service, Langerwehe, Germany) was used. The applied method was recently described by Höfele and Dürre 2023 [13].

#### 2.6.3. Gas Chromatography

The production of acetone and isopropanol was monitored via GC analysis. For this purpose, 2 mL of culture broth samples were taken during the growth of *A. woodii* and centrifuged (30 min at 4 °C and 13,000× *g*). The supernatant was transferred to glass vials (CS-Chromatographie Service GmbH, Langerwehe, Germany), acidified with 2 M HCl and sealed with screwcaps (CS-Chromatographie Service GmbH, Langerwehe, Germany). The acetone and isopropanol concentrations were analyzed with the gas chromatograph Perkin Elmer Clarus 680 GC system (Perkin Elmer, Waltham, MA, USA) equipped with an Elite-FFAP capillary column (Perkin Elmer, Waltham, MA, USA) with a length of 30 m, inner diameter of 0.32 mm, and a density film of 0.25 µM. An amount of 1 µL of the supernatant was injected with a split ratio of 20. Amounts of 45 mL min^−1^ H_2_ and 450 mL min^−1^ synthetic air served as the carrier gas. The separation of acetone and isopropanol was performed by using a temperature program (temperature held for 4 min at 50 °C, temperature ramp with 40 °C per minute to a maximum of 240 °C and temperature held at 240 °C for 3 min). Solvents were detected by a flame ionization detector (FID). External quantification standards were used for calibration.

## 3. Results

### 3.1. Isopropanol Production with Recombinant A. woodii Strains

For acetone production, the plasmids pJIR750_ac2t2 and pJIR750_ac3t3 were constructed. After transformation into competent *A. woodii* cells, heterotrophic and autotrophic growth experiments were performed. During the heterotrophic growth of *A. woodii* [pJIR750_ac2t2] and *A. woodii* [pJIR750_ac3t3] with 60 mM fructose as the substrate, 1.61 mM and 16.89 mM acetone together with 1.31 mM and 1.19 mM isopropanol were produced, respectively (data not shown). The autotrophic growth of *A. woodii* [pJIR750_ac2t2] and *A. woodii* [pJIR750_ac3t3] with CO_2_ + H_2_ as the substrate resulted in the production of 0.38 mM and 1.6 mM acetone. During the metabolism of CO_2_ + H_2_, no isopropanol was produced by the recombinant strains (Appendix A).

Heterologous isopropanol production was established by insertion of the *sadH* genes of *C. beijerinckii* DSM 6423, *C. beijerinckii* DSM 15410 and *C. ljungdahlii* into the acetone production plasmids pJIR750_ac1t1, pJIR750_ac1t2, pJIR750_ac2t1, pJIR750_ac2t2 and pJIR750_ac3t3 (Table 3) [16]. The 15 different plasmids for isopropanol production were transformed into electrocompetent *A. woodii* cells, and heterotrophic and autotrophic growth was performed and monitored.

#### 3.1.1. Isopropanol Production with the *sadH* Gene of *C. beijerinckii* DSM 6423

Five isopropanol production strains (*A. woodii* [pJIR750_ac1t1s1], *A. woodii* [pJIR750_ac1t2s1], *A. woodii* [pJIR750_ac2t1s1], *A. woodii* [pJIR750_ac2t2s1] and *A. woodii* [pjIR750_ac3t3s1]) were cultivated under heterotrophic conditions with 60 mM fructose (Figure 1A–C). During growth, a maximum of 10.75 ± 1.48 mM acetone was produced by *A. woodii* [pJIR750_ac2t1s1], while 8.56 ± 4.10 mM isopropanol were formed. Most isopropanol was produced by *A. woodii* [pJIR750_ac3t3s1] with 13.67 ± 3.58 mM and a production of 9.83 ± 0.42 mM acetone. Cultivation of *A. woodii* [pJIR750_ac1t1s1] and *A. woodii* [pJIR750_ac1t2s1] resulted in 4.94 ± 3.40 mM and 10.27 ± 2.20 mM acetone along with 1.41 ± 0.95 mM and 8.32 ± 4.99 mM isopropanol, respectively. *A. woodii* [pJIR750_ac2t2s1] reached concentrations of 10.10 ± 0.83 mM acetone and 9.63 ± 3.27 mM isopropanol. As fructose was consumed completely by all recombinant strains, a maximum OD_600_ of 3.3–3.5 was reached, while the pH dropped to 6.2–6.6. The natural product acetate was accumulated by the recombinant *A. woodii* strains in concentrations of 118–181 mM (Appendix A).

The production of isopropanol during autotrophic growth reached a maximum of 5.64 ± 1.08 mM by *A. woodii* [pJIR750_ac3t3s1], while producing 10.79 ± 2.25 mM acetone (Figure 1D–F). *A. woodii* [pJIR750_ac1t1s1] and *A. woodii* [pJIR750_ac1t2s1] produced 5.95 ± 1.02 mM and 3.38 ± 0.22 mM acetone and 0.76 ± 0.25 mM and 0.38 ± 0.09 mM isopropanol, respectively. The strains *A. woodii* [pJIR750_ac2t1s1] and *A. woodii* [pJIR750_ac2t2s1] reached a maximum acetone concentration of 7.72 ± 0.51 mM and 7.40 ± 1.23 mM and an isopropanol concentration of 0.49 ± 0.14 mM and 1.57 ± 0.77 mM, respectively. A maximum OD_600_ in the range of 1.45 to 1.68 was reached during consumption of 1.25 to 2.18 bar of CO_2_ + H_2_. During the production of 168–181 mM acetate, the pH of all strains dropped similarly (Appendix A). *A. woodii* wild-type and *A. woodii* [pJIR750] served as the control strains and showed no production of acetone and isopropanol.

#### 3.1.2. Isopropanol Production with the *sadH* Gene of *C. beijerinckii* DSM 15410

As the BLAST analysis revealed a 97% similarity of the Locus tag K684DRAFT_02545 in the genome of *C. beijerinckii* DSM 15410 to the *sadH* gene of *C. beijerinckii* DSM 6423, the *sadH* gene of *C. beijerinckii* DSM 15410 was inserted into the acetone production plasmids. The isopropanol and acetone production of the newly constructed strains *A. woodii* [pJIR750_ac1t1s2], *A. woodii* [pJIR750_ac1t2s2], *A. woodii* [pJIR750_ac2t1s2], *A. woodii* [pJIR750_ac2t2s2] and *A. woodii* [pJIR750_ac3t3s2] were examined by a heterotrophic growth experiment using 60 mM fructose as the carbon and energy source (Figure 2A–C). The highest isopropanol production was measured after the cultivation of *A. woodii* [pJIR750_ac3t3s2] with a concentration of 20.44 ± 0.16 mM in combination with a production of 9.22 ± 0.01 mM acetone. *A. woodii* [pJIR750_ac1t1s2] and *A. woodii* [pJIR750_ac1t2s2] reached concentrations of 5.89 ± 0.41 and 11.25 ± 0.59 mM acetone and 1.78 ± 0.31 and 15.27 ± 0.48 mM isopropanol, respectively. The strains *A. woodii* [pJIR750_ac2t1s2] and *A. woodii* [pJIR750_ac2t2s2] produced 3.45 ± 0.92 mM and 6.16 ± 1.28 mM acetone as well as 6.37 ± 0.43 mM and 10.90 ± 2.16 mM isopropanol, respectively. During growth, an OD_600_ of 3.3–3.9 was reached by the recombinant strains, and the production of acetate in concentrations of 84 mM to 145 mM was measured (Appendix A). Strains that produced less isopropanol and acetone showed a pH drop to 6.3 as stronger producers showed a pH of 6.9 to 7.1.

The autotrophic production of isopropanol was carried out using CO_2_ + H_2_ as the substrate, which resulted in a maximum isopropanol and acetone production of *A. woodii* [pJIR750_ac3t3s2] with 4.20 ± 0.69 mM and 10.30 ± 3.06 mM, respectively (Figure 2D–F). In total, 4.37 ± 1.41 mM and 2.80 ± 0.55 mM acetone together with 0.17 ± 0.14 mM and 0.83 ± 0.22 mM isopropanol were produced by *A. woodii* [pJIR750_ac1t1s2] and *A. woodii* [pJIR750_ac1t2s2], respectively. The strains *A. woodii* [pJIR750_ac2t1s2] and *A. woodii* [pJIR750_ac2t2s2] showed the autotrophic production of 6.45 ± 1.47 mM and 6.59 ± 0.97 mM acetone and 0.54 ± 0.02 and 1.53 ± 0.49 mM isopropanol. Due to an uptake of 1.7 to 2.2 bar of CO_2_ + H_2_, maximum OD_600_ values of 1.5 to 1.6 were reached by the recombinant strains. Acetate was produced in a concentration range of 112 mM to 174 mM and the pH dropped similarly for all strains (Appendix A).

#### 3.1.3. Isopropanol Production with the *sadH* Gene of *C. ljungdahlii*

The *sadH* gene of *C. ljungdahlii* demonstrated no significant similarity to the ones of *C. beijerinckii* DSM 6423 and DSM 15410, which was why it was chosen as a third candidate for isopropanol production in recombinant *A. woodii* strains. To examine isopropanol production, the strains *A. woodii* [pJIR750_ac1t1s3], *A. woodii* [pJIR750_ac1t2s3], *A. woodii* [pJIR750_ac2t1s3], *A. woodii* [pJIR750_ac2t2s3] and *A. woodii* [pJIR750_ac3t3s3] were cultivated heterotrophically with 60 mM fructose as the substrate (Figure 3A–C; Appendix A). The highest production of isopropanol was measured for *A. woodii* [pJIR750_ac3t3s3] with 18.06 ± 0.65 mM together with 11.17 ± 0.62 mM acetone. *A. woodii* [pJIR750_ac1t1s3] and *A. woodii* [pJIR750_ac1t2s3] produced 5.81 ± 2.81 mM and 10.51 ± 3.28 mM acetone as well as 6.77 ± 6.45 mM and 13.31 ± 7.11 mM isopropanol, respectively. The strains *A. woodii* [pJIR750_ac2t1s3] and *A. woodii* [pJIR750_ac2t2s3] reached acetone concentrations of 12.06 ± 1.86 mM and 10.66 ± 1.61 mM and isopropanol concentrations of 10.11 ± 3.71 mM and 9.99 ± 1.76 mM, respectively. While a maximum OD_600_ of 3.3–3.9 was reached, the pH dropped for all recombinant strains similarly.

During autotrophic growth with CO_2_ + H_2_ as the carbon and energy source, a maximum of 5.33 ± 0.37 mM isopropanol and 9.54 ± 0.41 mM acetone was produced by *A. woodii* [pJIR750_ac3t3s3] (Figure 3D–F; Appendix A). Isopropanol was also produced by *A. woodii* [pJIR750_ac1t1s3] and *A. woodii* [pJIR750_ac1t2s3] with a concentration of 0.26 ± 0.07 mM and 0.57 ± 0.16 mM together with 3.54 ± 0.12 mM and 1.84 ± 0.20 mM acetone. The strains *A. woodii* [pJIR750_ac2t1s3] and *A. woodii* [pJIR750_ac2t2s3] showed autotrophic production of 0.72 ± 0.10 mM and 4.27 ± 0.50 mM isopropanol in combination with 4.89 ± 0.29 mM and 11.58 ± 1.09 mM acetone. An uptake of 1.4–2.3 bar of CO_2_ + H_2_ resulted in maximum OD_600_ values of 1.5–1.9. The pH dropped similarly for all recombinant strains.

### 3.2. Improvement of Isopropanol Production in Recombinant A. woodii Strains

#### 3.2.1. Promotion of Isopropanol Production with the Gene *hydG* of *C. beijerinckii* DSM 6423

During fermentation of the recombinant *A. woodii* strains (described above), the heterologous production of isopropanol could be established. Despite expression of the *sadH* gene, encoding a secondary alcohol dehydrogenase, acetone was not completely converted to isopropanol. To promote the conversion of acetone to isopropanol, the *hydG* gene, which is annotated as a gene for a potential electron donor protein [20], should be included in the best-performing isopropanol plasmids. After several unsuccessful trials to insert *hydG* into the plasmid pJIR750_ac3t3s1, it was decided to construct a two-plasmid system. Therefore, the gene *hydG* of *C. beijerinckii* DSM 6423 was inserted into the vector pMTL83251 under the control of the P*_thlA_* promoter. The resulting plasmid pMTL83251_PthlA_h1 was transformed into electrocompetent *A. woodii* [pJIR750_ac3t3s1], *A. woodii* [pJIR750_ac3t3s2] and *A. woodii* [pJIR750_ac3t3s3] cells, and isopropanol production was examined by performing a heterotrophic growth experiment with 60 mM fructose (Figure 4; Appendix A). The control strains *A. woodii* [pJIR750_ac3t3s1], *A. woodii* [pJIR750_ac3t3s2] and *A. woodii* [pJIR750_ac3t3s3] produced maximum isopropanol concentrations of 17.51 ± 1.88 mM, 16.39 ± 1.25 mM and 17.91 ± 4.39 mM, respectively. In comparison, the strains *A. woodii* [pJIR750_ac3t3s1] [pMTL83251_PthlA_h1], *A. woodii* [pJIR750_ac3t3s2] [pMTL83251_PthlA_h1] and *A. woodii* [pJIR750_ac3t3s3] [pMTL83251_PthlA_h1] reached maximum isopropanol concentrations of 8.11 ± 0.65 mM, 13.08 ± 3.14 mM and 9.17 ± 2.22 mM, respectively. The production of acetone was even less for strains harboring both plasmids. Acetone concentrations of 8.05 ± 0.21 mM, 8.21 ± 1.82 mM and 7.20 ± 1.85 mM were monitored for *A. woodii* [pJIR750_ac3t3s1] [pMTL83251_PthlA_h1], *A. woodii* [pJIR750_ac3t3s2] [pMTL83251_PthlA_h1] and *A. woodii* [pJIR750_ac3t3s3] [pMTL83251_PthlA_h1], respectively. However, *A. woodii* [pJIR750_ac3t3s1], *A. woodii* [pJIR750_ac3t3s2] and *A. woodii* [pJIR750_ac3t3s3] produced maximum acetone concentrations of 9.65 ± 1.27 mM, 10.80 ± 0.97 mM and 10.35 ± 2.35 mM, respectively. As the insertion of *hydG* did not lead to higher isopropanol production and better conversion rates, autotrophic growth was not investigated.

#### 3.2.2. Overcoming the Bottlenecks of Recombinant Isopropanol Production

As the additional insertion of *hydG* originating from *C. beijerinckii* DSM 6423 did not lead to the enhanced conversion of acetone to isopropanol, but to a decrease in total solvent production, the location of *hydG* in the genome and its function was further analyzed. Thereby, it was found that *hydG* harbors a σ^54^ interaction domain, indicating that it serves as an σ^54^ activator protein [26]. Further analysis of the intergenic region between *sadH* and *hydG* revealed the typical −24/−12 consensus recognition sequence for σ^54^ binding [27]. With this background of Wang et al. [18], showing enhanced isopropanol production by insertion of the *sadH*-*hydG* gene cluster, the cluster was inserted into the vector pMTL83251. The resulting plasmid pMTL83251_PthlA_sh1 was transformed into electrocompetent cells of the acetone producer *A. woodii* [pJIR750_ac3t3] and the isopropanol producer *A. woodii* [pJIR750_ac3t3s1].

Additionally, another bottleneck in the isopropanol production pathway was addressed, which is found in the conversion efficiency of the CoA-transferase [2]. To bypass the bottleneck, an additional gene copy of *ctfA/ctfB* originating from *C. scatologenes* was inserted into pMTL83251_PthlA_sh1. The constructed plasmid pMTL83251_PthlA_sh1c3 was also transformed into the recombinant strains *A. woodii* [pJIR750_ac3t3] and *A. woodii* [pJIR750_ac3t3s1].

The enhanced conversion of acetone to isopropanol was examined by the performance of both a heterotrophic and an autotrophic growth experiment with the recombinant strains *A. woodii* [pJIR750_ac3t3] [pMTL83251_PthlA_sh1], *A. woodii* [pJIR750_ac3t3] [pMTL83251_PthlA_sh1c3], *A. woodii* [pJIR750_ac3t3s1] [pMTL83251_PthlA_sh1] and *A. woodii* [pJIR750_ac3t3s1] [pMTL83251_PthlA_sh1c3] with 60 mM fructose or CO_2_ + H_2_ as the substrates, respectively (Figure 5; Appendix A). *A. woodii* wild-type, *A. woodii* [pJIR750], *A. woodii* [pJIR750] [pMTL83251], *A. woodii* [pJIR750_ac3t3] and *A. woodii* [pJIR750_ac3t3s1] were cultivated as the control strains. During heterotrophic growth, *A. woodii* [pJIR750_ac3t3] [pMTL83251_PthlA_sh1] and *A. woodii* [pJIR750_ac3t3] [pMTL83251_PthlA_sh1c3] reached a maximum isopropanol production of 15.30 ± 0.86 mM and 17.50 ± 1.39 mM, respectively. Acetone was monitored as a byproduct for both strains with a maximum concentration of 3.84 ± 0.31 mM and 4.86 ± 0.25 mM, respectively. In comparison, the control strain *A. woodii* [pJIR750_ac3t3] showed a maximum production of 2.23 ± 0.12 mM of isopropanol and 20.31 ± 0.89 mM of acetone, as the strain is lacking a plasmid containing the *sadH* gene. The recombinant strains *A. woodii* [pJIR750_ac3t3s1] [pMTL83251_PthlA_sh1] and *A. woodii* [pJIR750_ac3t3s1] [pMTL83251_PthlA_sh1c3] produced a maximum of 16.33 ± 0.50 mM and 16.18 ± 0.78 mM isopropanol, together with 5.44 ± 0.40 mM and 6.08 ± 0.45 mM acetone, respectively. *A. woodii* [pJIR750_ac3t3s1] served as a control strain and reached a maximum isopropanol concentration of 17.79 ± 3.48 mM in combination with 7.09 ± 0.42 mM acetone. The control strains *A. woodii* wild-type, *A. woodii* [pJIR750] and *A. woodii* [pJIR750] [pMTL83251] showed no production of isopropanol or acetone. All strains reached a maximum OD_600_ of 1.9–3.2. The recombinant isopropanol and acetone producer showed a pH drop to 6.5–6.6. In comparison, the pH of the control strains *A. woodii* wild-type, *A. woodii* [pJIR750] and *A. woodii* [pJIR750] [pMTL83251] decreased to 5.8.

During autotrophic growth using CO_2_ + H_2_ as the carbon and energy source, the strain *A. woodii* [pJIR750_ac3t3] [pMTL83251_PthlA_sh1] as well as the control strain *A. woodii* [pJIR750_ac3t3] showed highly impaired growth, whereas *A. woodii* [pJIR750_ac3t3] [pMTL83251_PthlA_sh1c3] did not grow at all. Accordingly, acetone and isopropanol concentrations of *A. woodii* [pJIR750_ac3t3] [pMTL83251_PthlA_sh1] were below the quantification limit. The control strain *A. woodii* [pJIR750_ac3t3] produced 1.66 ± 1.04 mM of acetone and no isopropanol. The recombinant isopropanol producers harboring two copies of the *sadH* gene grew similarly to the control strains *A. woodii* wild-type, *A. woodii* [pJIR750] and *A. woodii* [pJIR750] [pMTL83251] to a maximum OD_600_ of 1.5–1.8. *A. woodii* [pJIR750_ac3t3s1] [pMTL83251_PthlA_sh1] and *A. woodii* [pJIR750_ac3t3s1] [pMTL83251_PthlA_sh1c3] reached maximum isopropanol concentrations of 7.02 ± 0.44 mM and 13.87 ± 1.97 mM, together with acetone concentrations of 0.65 ± 0.20 mM and 1.35 ± 0.14 mM, respectively. In comparison, the control strain *A. woodii* [pJIR750_ac3t3s1] produced 3.26 ± 0.30 mM of isopropanol and 3.19 ± 0.59 mM of acetone.

## 4. Discussion

The discovery of how to expend fossil fuels, such as natural gas, petroleum or coal, has played a severe role in the changes to human society since the late 18th century. Due to the use of fossil fuels, the standards of living, wealth and human health could be improved and led us to the way of today’s living [28]. Besides the undeniable positive effect of industrialization and use of fossil fuels on society, there are also crucial disadvantages and threats coming up with the utilization of fossil resources. As the burning of fossil fuels comes together with the emission of CO_2_, it drives the greenhouse gas effect and accelerates global warming [29]. The estimated global warming of 1–5 °C during the next century will generate environmental changes, e.g., melting of glaciers, rising sea levels and extreme weather events, with serious impacts on flora, fauna and human life [30,31]. Additionally, fossil resources are limited and will be depleted in a few years if today’s consumption will continue. To counteract this phenomenon and retard global warming, alternatives for fossil-based processes should be developed and investigated. A promising alternative, which has gained interest throughout the last decades, is the production of high-value chemicals by fermentation using acetogens [32,33]. The exchange of fossil-based procedures against acetogenic fermentation comes up with two major advantages: Fossil resources are saved since acetogens are capable of metabolization of single carbon gases (C_1_-gases). Further, the emission of C_1_-gases that accelerate global warming can be diminished by the production of high-value chemicals using acetogens and result in the best case in a carbon-neutral way of manufacturing [2]. Since acetogenic bacteria are defined as a class of microorganisms harboring the enzymes of the WLP, members of acetogens are widely spread throughout many genera [6]. Thereby, the natural product spectrum of acetogens is versatile, comprising, e.g., ethanol, 2,3-butanediol or hexanol [34,35]. Further, the development of toolboxes for metabolic engineering expands the economic potential of some acetogens, which are genetically accessible. One model organism with established protocols for metabolic engineering and recently also genomic editing via its endogenous CRISPR/Cas system is *A. woodii* [15,36,37]. The natural product spectrum of *A. woodii* comprises solely acetate and traces of ethanol under certain circumstances [38,39]. By successful metabolic engineering, *A. woodii* was recently employed to produce lactate, poly-3-hydroxybutyrate or acetone together with the unexpected accumulation of isopropanol [13,14,15,16].

The reduction of acetone to isopropanol is also known in some natural isopropanol-producing *Clostridia* species, e.g., *C. beijerinckii* and *C. aurantibutyricum*; the latter was recently reclassified as *C. felsineum* [40,41]. *Clostridium* species capable of isopropanol production harbor a primary/secondary alcohol dehydrogenase (SadH) specific for acetone, which is strictly NADPH-dependent [17]. The production of isopropanol occurs in clostridial species in combination with butanol and ethanol and is a modification of the acetone–butanol–ethanol (ABE) fermentation. A prominent candidate for ABE fermentation is *C. acetobutylicum*, which was formerly employed for acetone production during World War I [42]. For *C. beijerinckii* species, butanol is the favored product, whereby the production of isopropanol at an industrial scale reveals that it is unprofitable. Therefore, recombinant isopropanol production was examined using *E. coli*, resulting in a maximum production of up to 40 g/L [43]. Due to enormous high solvent titers and the biocidal properties of isopropanol, cost-intensive gas stripping for the removal of produced isopropanol was necessary. Furthermore, fermentation using glucose as a substrate proved to be expensive, which is why researchers focused on the production of high-value chemicals using cheap waste gases as feedstock. Efficient and economically competitive processes have been recently developed by LanzaTech, which are using genomically modified *C. autoethanogenum* strains for the production of ethanol based on synthesis gas (syngas; a mixture of H_2_, CO_2_ and CO) [8,9]. LanzaTech is also working on the economically profitable production of isopropanol with recombinant *C. autoethanogenum* using syngas as the substrate. By the screening of a huge industrial genome databank and use of kinetic models, Liew et al. constructed a strain capable of producing approximately 3 g/L/h of isopropanol with syngas as the feedstock. Therefore, the native *sadH* gene of *C. autoethanogenum* was employed [2].

During the recombinant production of acetone with *A. woodii*, traces of isopropanol could be detected. Since the analysis of the genome of *A. woodii* regarding a *sadH* gene, responsible for the conversion of acetone to isopropanol, was not successful until now, this study focused on the heterologous production of isopropanol [16]. Therefore, the plasmid-based expression of the required genes *thlA* (encoding a thiolase A), *ctfA/ctfB* (encoding a CoA-transferase), *adc* (encoding an acetoacetate decarboxylase) and *sadH* (encoding a secondary alcohol dehydrogenase) was established (Figure 6). To find the best recombinant isopropanol producer, a library with genes originating from different *Clostridium* species was established. As acetone production in *A. woodii* was first examined using *thlA*, *ctfA/ctfB* and *adc* of *C. acetobutylicum*, those genes were also chosen for isopropanol production [15]. Due to a high K_m_ value of *ctfA/ctfB* originating from *C. acetobutylicum*, the genes of *C. aceticum*, *C. kluyveri* and *C. scatologenes* were also tested [16,44]. Variation of the gene encoding the thiolase A proved to be promising for acetone production, which is why it was also examined regarding isopropanol accumulation in this study. The gene library was completed by the *sadH* genes of *C. beijerinckii* and *C. ljungdahlii* [17,18,19]. Since isopropanol was an unexpected byproduct of recombinant acetone production in *C. autethanogenum*, the responsible gene of *C. autoethanogenum* was also taken into consideration [2]. However, BLAST analysis revealed a 100% similarity of the respective gene to the *sadH* of *C. ljungdahlii*. The same result was found for comparison with the genome of *C. coskatii*. Genome analysis of *C. beijerinckii* DSM 15410, which was formerly known as *C. diolis*, revealed a candidate gene coding for a secondary alcohol dehydrogenase with 97% identity to the *sadH* of *C. beijerinckii* DSM 6423 [45].

The combination of the acetone production genes of *C. acetobutylicum* with the candidate genes for *sadH* in recombinant *A. woodii* confirmed that autotrophic production of isopropanol is possible. As productions levels ranged between 0.17 and 0.76 mM, different gene combinations were examined. Thereby, it was found that any plasmid containing the *thlA* or *ctfA/ctfB* genes originating from *C. acetobutylicum* resulted in similar isopropanol concentrations during autotrophic growth (Table 4). Higher isopropanol accumulation was reached by exchanging *thlA* and *ctfA/ctfB* against genes originating from *C. kluyveri* and *C. aceticum*, respectively. The combination of those genes with the *sadH* of *C. beijerinckii* DSM 6423 and DSM 15410 resulted in two-fold increased isopropanol concentrations of 1.53–1.57 mM. By the expression of the SadH originating from *C. ljungdahlii*, the isopropanol production was increased to 4.27 mM. The most promising recombinant isopropanol producers were found using the *thlA* and *ctfA/ctfB* of *C. scatologenes*. Combining those genes with the *sadH* of *C. ljungdahlii* and *C. beijerinckii* DSM 6423, the isopropanol concentration could be raised up to 5.33–5.64 mM.

Since isopropanol production was already established in *C. acetobutylicum* and *C. saccharoperbutylacetonicum*, as well as in the acetogens *C. ljungdahlii* and *C. autoethanogenum*, bottlenecks of the production pathway were already discovered [2,19,20,21]. One limitation was found in the reduction efficiency of acetone to isopropanol by the SadH, indicated by leftover acetone concentrations. In case of fermentation using recombinant *C. saccharoperbutylacetonicum*, the conversion efficiency was increased from 50% to approximately 95% by additionally expressing the gene *hydG,* which is located in the same operon as *sadH*. Wang et al. [20] hypothesized that *hydG* encodes an electron transfer protein, delivering electrons to raise the NADPH supply necessary for the reduction of acetone. A positive effect by the co-expression of *hydG* and *sadH* was only detected if the *sadH*–*hydG* operon was integrated into the genome of *C. saccharoperbutylacetonicum*. Plasmid-based expression of the operon did not lead to enhanced acetone reduction [20]. The co-expression of *hydG* and *sadH* was also tested in this study to improve isopropanol concentrations and reduce acetone accumulation. Since the construction of a plasmid harboring all enzymes required for isopropanol production and *hydG* was not successful, a two-plasmid system was established. Therefore, a plasmid harboring the gene *hydG* under the control of the P*_thlA_* promoter was constructed. The effect of the newly constructed plasmid was investigated using the best performing isopropanol producers of previous experiments. Thereby, recombinant strains harboring two plasmids showed both a significantly inferior isopropanol and also total solvent production. Further analysis of HydG exposed several σ^54^ interaction domains, indicating that the protein acts as a σ^54^-dependent transcription activator. The sigma factor σ^54^ is involved in the regulation of gene transcription via binding to the RNA polymerase. The σ^54^ holoenzyme is not sufficient to bind to the RNA polymerase, but it requires an activator protein, which is bound by the hydrolysis of ATP. The transcription regulating complex binds to the conserved −24 (GG)/−12 (GC) promoter sequence, which was found in the intergenic region between *hydG* and *sadH* in the genome of *C. beijerinckii* DSM 6423 [26,27]. For that reason, a new plasmid was constructed harboring the *sadH*–*hydG* operon including the σ^54^-binding promoter sequence in the intergenic region. The effect of the newly constructed plasmid was examined in combination with the acetone production plasmid pJIR750_ac3t3 to maintain the copy number of *sadH*. In comparison to isopropanol production with *A. woodii* [pJIR750_ac3t3s1], the co-expression of both plasmids resulted in similar isopropanol accumulation. Based on those results, it remains unclear whether active HydG is expressed. It is also not known if σ^54^ is expressed in *A. woodii*, although the gene encoding the sigma factor is present in the genome. Transcriptomic analysis or fusion of *hydG* to the fluorescence-activating and absorption-shifting tag (FAST) for expression control would be clarifying [46]. As acetogens are living at the thermodynamic limit of life, ATP restriction could also hinder the ATP-dependent binding of *hydG* to σ^54^. A second solution to enhance acetone reduction is the multiplication of *sadH* gene copy numbers. With this approach, isopropanol production could be successfully increased to 7.02 ± 0.44 mM.

A second bottleneck discovered in the isopropanol production pathway is the efficiency of the CoA-transferase. Via examination of the *ctfA/ctfB* and *thlA* genes originating from different bacterial species, isopropanol production could be increased from an initial 0.76 ± 0.25 mM to a maximum 5.64 ± 1.08 mM, which means a 7.5-fold improvement. For the recombinant production of isopropanol using metabolically engineered *C. autoethanogenum*, insertion of a second gene copy of *ctfA/ctfB* led to significantly raised isopropanol levels [2]. The same approach was considered for improving isopropanol production using *A. woodii*. The best performing *A. woodii* strain, harboring two gene copies of both *sadH* and *ctfA/ctfB* produced 13.87 ± 1.97 mM (0.83 g/L) of isopropanol, indicating an increase by more than 18-fold compared to the initial isopropanol accumulation. This is still only a third of the production ability of *C. autoethanogenum*, but provides so far the most successful example of isopropanol formation from CO_2_ + H_2_ under anaerobic conditions. It can be summarized that the heterologous production of isopropanol using recombinant *A. woodii* strains is possible. Variation in gene origin as well as the addition of multiple gene copies can help to improve the isopropanol yield. In this case, gene variation and the doubling of the genes *sadH* and *ctfA/ctfB* increased the initial isopropanol concentration of 0.76 ± 0.25 mM by 18-fold to 13.87 ± 1.97 mM.

To reveal the economic potential of the newly constructed isopropanol producing *A. woodii* strain, upscaling of the process via continuous fermentation would be beneficial. Besides fermentation using syngas, the microbial production of isopropanol using CO_2_ + H_2_ could be another promising alternative for the petroleum-based production of high-value chemicals.

## Figures and Tables

**Figure 1 bioengineering-10-01381-f001:**
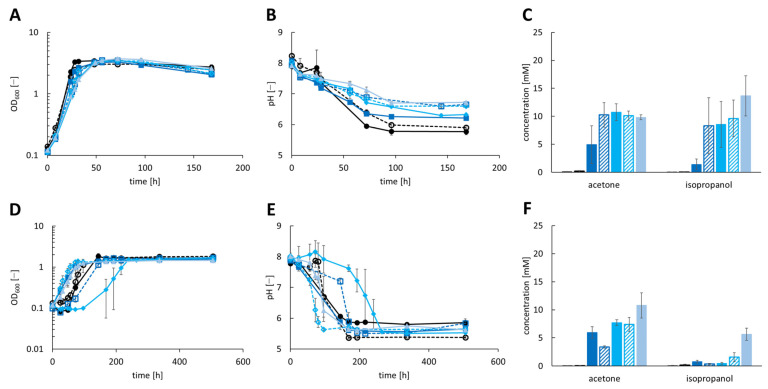
Production of isopropanol using recombinant *A. woodii* strains, harboring the secondary alcohol dehydrogenase (*sadH*) gene of *C. beijerinckii* DSM 6423, cultivated with 60 mM fructose (**A**–**C**) or CO_2_ + H_2_ (**D**–**F**) as substrate. Growth was performed in biological triplicates with *A. woodii* wild-type and *A. woodii* [pJIR750] as control strains (depicted as black line with filled circles or filled black column and dashed black line with empty circles or dashed black column, respectively). During growth, (**A**,**D**) the optical density at 600 nm (OD_600_), (**B**,**E**) changes in pH and (**C**,**F**) product concentrations of acetone and isopropanol were monitored. The recombinant strains *A. woodii* [pJIR750_ac1t1s1] and *A. woodii* [pJIR750_ac1t2s1] are depicted as dark blue lines with filled rectangles or filled dark blue column and dashed dark blue lines with empty rectangles or dashed dark blue column, respectively. *A. woodii* [pJIR750_ac2t1s1] and *A. woodii* [pJIR750_ac2t2s1] are shown as light blue lines with filled rhombus or light blue column and dashed light blue lines with empty rhombus or dashed light blue column, respectively. The pastel blue lines with filled triangle or pastel blue column depict *A. woodii* [pJIR750_ac3t3s1].

**Figure 2 bioengineering-10-01381-f002:**
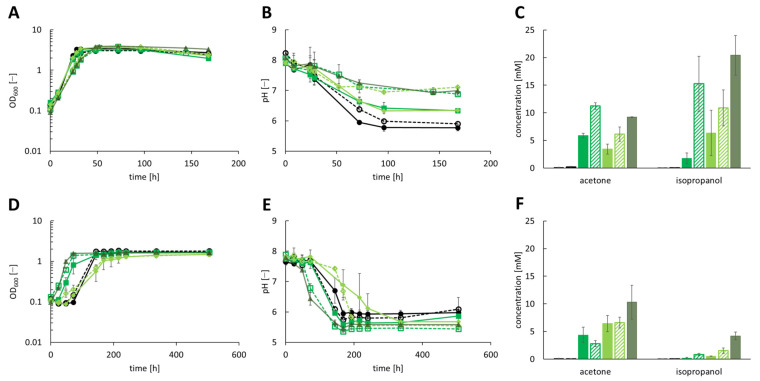
Production of isopropanol using recombinant *A. woodii* strains, harboring the *sadH* gene of *C. beijerinckii* DSM 15410, cultivated with 60 mM fructose (**A**–**C**) or CO_2_ + H_2_ (**D**–**F**) as substrate. Growth was performed in biological triplicates with *A. woodii* wild-type and *A. woodii* [pJIR750] as control strains (depicted as black line with filled circles or filled black column and dashed black line with empty circles or dashed black column, respectively). During growth, (**A**,**D**) OD_600_, (**B**,**E**) changes in pH and (**C**,**F**) product concentrations of acetone and isopropanol were monitored. The recombinant strains *A. woodii* [pJIR750_ac1t1s2] and *A. woodii* [pJIR750_ac1t2s2] are depicted as green lines with filled rectangles or filled green column and dashed green lines with empty rectangles or dashed green column, respectively. *A. woodii* [pJIR750_ac2t1s2] and *A. woodii* [pJIR750_ac2t2s2] are shown as light green lines with filled rhombus or filled light green column and dashed light green lines with empty rhombus or dashed light green column, respectively. The brown-green lines with filled triangle or brown-green column depict *A. woodii* [pJIR750_ac3t3s2].

**Figure 3 bioengineering-10-01381-f003:**
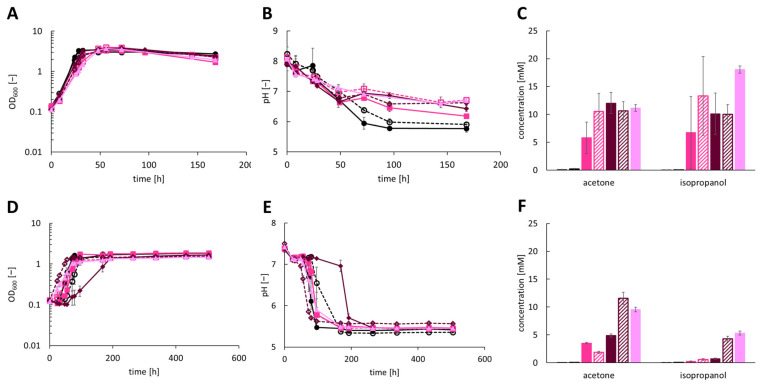
Production of isopropanol using recombinant *A. woodii* strains, harboring the *sadH* gene of *C. ljungdahlii*, cultivated with 60 mM fructose (**A**–**C**) or CO_2_ + H_2_ (**D**–**F**) as substrate. Growth was performed in biological triplicates with *A. woodii* wild-type and *A. woodii* [pJIR750] as control strains (depicted as black line with filled circles or filled black column and dashed black line with empty circles or dashed black column, respectively). During growth, (**A**,**D**) OD_600_, (**B**,**E**) changes in pH and (**C**,**F**) product concentrations of acetone and isopropanol were monitored. The recombinant strains *A. woodii* [pJIR750_ac1t1s3] and *A. woodii* [pJIR750_ac1t2s3] are depicted as pink lines with filled rectangles or filled pink column and dashed pink lines with empty rectangles or dashed pink column, respectively. *A. woodii* [pJIR750_ac2t1s3] and *A. woodii* [pJIR750_ac2t2s3] are shown as purple lines with filled rhombus or filled purple column and dashed purple lines with empty rhombus or dashed purple column, respectively. The rosa lines with filled triangle or rosa column depict *A. woodii* [pJIR750_ac3t3s3].

**Figure 4 bioengineering-10-01381-f004:**
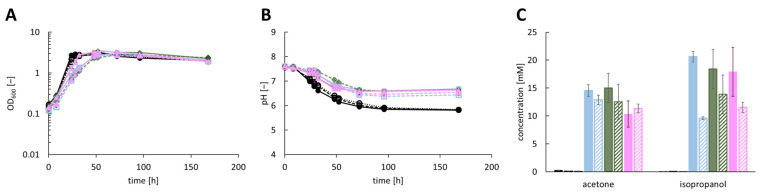
Production of isopropanol using recombinant *A. woodii* strains, harboring the *hydG* gene of *C. beijerinckii* DSM 6423, cultivated with 60 mM fructose as substrate. Growth was performed in biological triplicates with *A. woodii* wild-type, *A. woodii* [pJIR750] and *A. woodii* [pJIR750] [pMTL83251] as control strains (depicted as black line with filled circles or filled black column, dashed black line or column and dotted black line or column with empty circles, respectively). During growth, (**A**) OD_600_, (**B**) changes in pH and (**C**) maximum product concentrations of acetone and isopropanol were monitored. The recombinant strains *A. woodii* [pJIR750_ac3t3s1] and *A. woodii* [pJIR750_ac3t3s1] [pMTL83251_PthlA_h1] are depicted as pastel blue lines with filled rectangles or filled pastel blue columns and dashed pastel blue lines with empty rectangles or dashed pastel blue columns, respectively. *A. woodii* [pJIR750_ac3t3s2] and *A. woodii* [pJIR750_ac3t3s2] [pMTL83251_PthlA_h1] are shown as brown-green lines with filled rhombus or filled brown-green columns and dashed brown-green lines with empty rhombus or dashed brown-green column, respectively. The rosa lines with filled triangle or filled rosa columns and dashed rosa lines with empty rectangles or dashed rosa columns depict *A. woodii* [pJIR750_ac3t3s3] and *A. woodii* [pJIR750_ac3t3s3] [pMTL83251_PthlA_h1], respectively.

**Figure 5 bioengineering-10-01381-f005:**
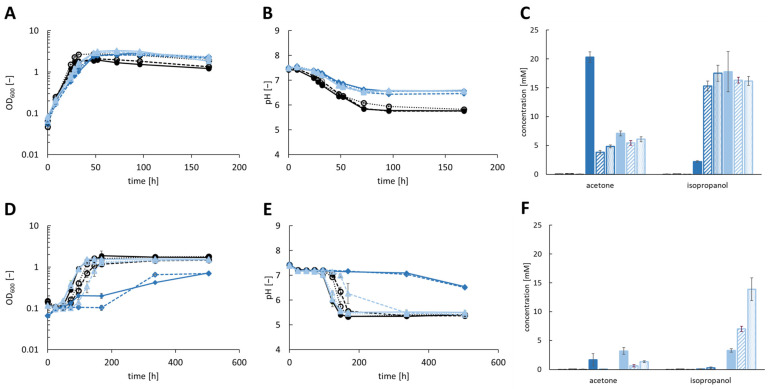
Production of isopropanol using recombinant *A. woodii* strains, harboring the *sadH*-*hydG* gene cluster of *C. beijerinckii* DSM 6423 and an additional copy of *ctfA/ctfB* originating from *C. scatologenes*, cultivated with 60 mM fructose (**A**–**C**) or CO_2_ + H_2_ (**D**–**F**) as substrate. Growth was performed in biological triplicates with *A. woodii* wild-type, *A. woodii* [pJIR750] and *A. woodii* [pJIR750] [pMTL83251] as control strains (depicted as black line with filled circles or filled black column, dashed black line or column and dotted black line or column with empty circles, respectively). During growth, (**A**,**D**) OD_600_, (**B**,**E**) changes in pH and (**C**,**F**) maximum product concentrations of acetone and isopropanol were monitored. The recombinant strains *A. woodii* [pJIR750_ac3t3], *A. woodii* [pJIR750_ac3t3] [pMTL83251_PthlA_sh1] and *A. woodii* [pJIR750_ac3t3] [pMTL83251_PthlA_sh1c3] are depicted as dark blue lines or column with filled rhombus, dashed dark blue lines or column with empty rhombus and dotted dark blue lines or column with empty rhombus, respectively. *A. woodii* [pJIR750_ac3t3s1], *A. woodii* [pJIR750_ac3t3s1] [pMTL83251_PthlA_sh1] and *A. woodii* [pJIR750_ac3t3s1] [pMTL83251_PthlA_sh1c3] are shown as pastel blue lines or column with filled triangles, dashed pastel blue lines or column with empty triangles and dotted pastel blue lines or column with empty triangles, respectively.

**Figure 6 bioengineering-10-01381-f006:**
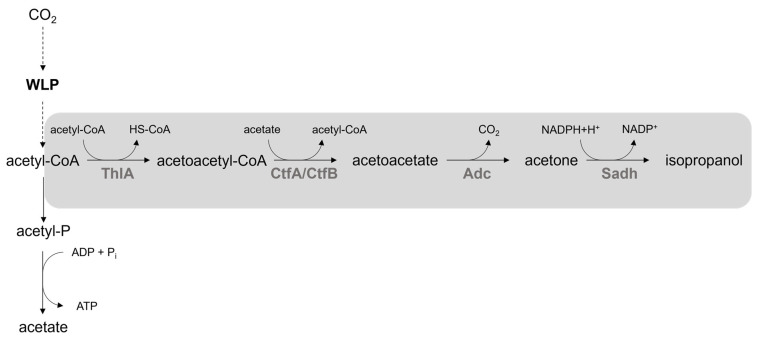
Recombinant isopropanol production pathway in *A. woodii*. Abbreviations: CoA, coenzyme A; NADP^+^, nicotinamide adenine dinucleotide phosphate; H^+^, proton; P, phosphate; WLP, Wood–Ljungdahl pathway; ThlA, thiolase A; CtfA/CtfB, CoA-transferase; Adc, acetoacetate decarboxylase; Sadh, secondary alcohol dehydrogenase.

**Table 1 bioengineering-10-01381-t001:** Bacterial strains, their genotype and origin used in this work.

Strain	Genotype	Origin
*Escherichia coli* XL1-Blue MRF’	Δ (*mcrA*)183 Δ (*mcrCB-hsd SMR-mrr*)173 *endA1 supE44 thi-1recA1 gyrA96 relA1* *lac* [*F’proAB lacIq* *Z*Δ*M15 Tn10* (*TetR*)]	Agilent Technologies (Santa Clara, CA, USA)
*Acetobacterium woodii* DSM 1030	Type strain	German Collection of Microorganisms and Cell Cultures (DSMZ, Brunswick, Germany)
*Clostridium kluyveri* DSM 555	Type strain	German Collection of Microorganisms and Cell Cultures (DSMZ, Brunswick, Germany)
*C. scatologenes* DSM 757	Type strain	German Collection of Microorganisms and Cell Cultures (DSMZ, Brunswick, Germany)
*C. beijerinckii* DSM 6423	Type strain	German Collection of Microorganisms and Cell Cultures (DSMZ, Brunswick, Germany)
*C. beijerinckii* DSM 15410	Type strain	German Collection of Microorganisms and Cell Cultures (DSMZ, Brunswick, Germany)
*C. ljungdahlii* DSM 13528	Type strain	German Collection of Microorganisms and Cell Cultures (DSMZ, Brunswick, Germany)

**Table 3 bioengineering-10-01381-t003:** Designation scheme for the genes of the isopropanol production plasmids according to their origin.

Plasmid	*ctfA/ctfB*	*thlA*	*sadH*
pJIR750_ac1t1s1	c1. *C. acetobutylicum*	t1. *C. acetobutylicum*	s1. *C. beijerinckii* DSM 6423
pJIR750_ac2t2s2	c2. *C. aceticum*	t2. *C. kluyveri*	s2. *C. beijerinckii* DSM 15410
pJIR750_ac3t3s3	c3. *C. scatologenes*	t3. *C. scatologenes*	s3. *C. ljungdahlii*

* a is used as abbreviation for *adc* originating from *C. acetobutylicum*.

**Table 4 bioengineering-10-01381-t004:** Summary and comparison of autotrophic isopropanol production [mM] of recombinant *A. woodii* strains. Abbreviations: s1, *sadH* gene of *C. beijerinckii* DSM 6423; s2, *sadH* gene of *C. beijerinckii* DSM 15410; s3, *sadH* gene of *C. ljungdahlii*.

Backbone	s1	s2	s3
pJIR750_ac1t1	0.76 ± 0.25	0.17 ± 0.14	0.26 ± 0.07
pJIR750_ac1t2	0.38 ± 0.09	0.83 ± 0.22	0.57 ± 0.16
pJIR750_ac2t1	0.49 ± 0.14	0.54 ± 0.02	0.72 ± 0.10
pJIR750_ac2t2	1.57 ± 0.77	1.53 ± 0.49	4.27 ± 0.50
pJIR750_ac3t3	5.64 ± 1.08	4.20 ± 0.69	5.33 ± 0.37

## Data Availability

The data presented in this study are available in “Heterologous production of isopropanol using metabolically engineered *Acetobacterium woodii* strains”.

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
