# Peer review of "Heterologous Production of Isopropanol Using Metabolically Engineered Acetobacterium woodii Strains"

_bioengineering, 2023, doi:10.3390/bioengineering10121381_

Round 1

Reviewer 1 Report

Comments and Suggestions for Authors

The manuscript “Heterologous production of isopropanol using metabolically engineered Acetobacterium woodii strains” by Franziska Höfele, Teresa Schoch, Catarina Oberlies and Peter Dürre explores microbial fermentation with acetogenic bacteria, specifically Acetobacterium woodii, to extend acetone production towards isopropanol. The researchers employed genetic modifications, introducing genes from several related organisms to optimize the isopropanol production pathway and address some of the known bottlenecks in the production process. Overall, this is an excellent paper presenting very solid research on a non-trivial system.

The experimental approach is rational and conclusive. The controls and replicates are appropriate and suffice highest scientific standards. The results are presented clearly with figures of high quality, and the chain of thought is easy to follow. Apart from a few very minor and almost nitpicky comments I have only one major concern, which pertains to the referencing of a flawed life-cycle analysis on biotechnological production of acetone and isopropanol (see comment to Line 482).

Other general but minor concern is that yields were not determined (see comment to line 303). Lastly, I am wondering if a correlation between OD and cell dry-weight was determined for A. woodii. This is, in general, not difficult (although would likely have to be done for fructose and CO2+H2 individually) and would add some tangibility to the whole study.

My further specific comments and suggestions follow below:

Line 16: “to this end” would be a better phrase than “therefore” in this context

Line 32: “therefore” would be a better phrase than “that is why” in this context

Line 40: “hence” would be a better word than “why” in this context

Line 52: “formation” would be a better word than “production” in this context

Line 58: “represents” would be a better word than “builds” in this context

Line 168: “Woodii” should be lower case

Line 182: “…, the gas-phase was replenished…”

Line 192: the formulation here is a bit cumbersome, I suggest changing as per the following: “The growth of A. woodii was monitored by means of the optical density (OD600: absorption of culture at 600 nm) and pH, measured as recently described.”

Line 193: for better accessibility to the paper it would be good if a brief description of the method (even if given in detail in the respective reference) was provided here

Line 197: “for this purpose” would be a better phrase than “therefore” in this context

Line 232: the formulation here is a bit cumbersome, I suggest changing as per the following: “Five isopropanol production strains (…) were cultivated under heterotrophic conditions with 60 mM fructose.”

Line 303: in this context, an improvement could be to determine the total uptake of CO2 for the purpose of estimating yields throughout the different experiments – suggestion only, I understand if this goes too far or if confidence in the data regarding substrate-consumption is not high enough

Line 348: for consistency with the other chapters, it would be good to indicate in this heading the originating organism of the hydG gene

Line 436: it would be nice if the authors could speak to their hypothesis for this impairment of growth in the discussion – could protein toxicity of the CoA-transferase be a potential reason?

Line 465: “crucial” would be a better word here than “severe”

Line 469: “threats” (not “threads”)

Line 475: really, it is already too late for that, what is needed is an actual reversal and carbon-capture to reduce atmospheric levels of carbon dioxide (that is not to say that emission-mitigation isn’t a compulsory part of the way towards a circular and sustainable biochemical manufacturing industry)

Line 482: This is, unfortunately, not true, and must be differentiated better if not to add to the greenwashing slide: using acetogens to convert carbon dioxide into fuels can, at best, be carbon-neutral (despite of what LanzaTech and collaborators wrongfully claim in 10.1038/s41587-021-01195-w) – it really depends on the origin of the feedstock: if the carbon is obtained through carbon-capture (e.g. from the atmosphere) the process could, theoretically, be carbon-neutral (if the products are fuels or other compounds with short life-cycles) or carbon-negative (only in a case where the carbon is locked away permanently); if emissions are utilized (such as e.g. steel-mill off-gas), the process can, at best be carbon-neutral (in case of products with significant durability), but in any other case the process will still be carbon-positive, as the feedstocks are ultimately still derived from fossil fuels. The referenced study (10.1038/s41587-021-01195-w) does not take that into account, as their “cradle-to-gate” life-cycle analysis only considers a narrow section of the whole process. In order to not further promote these misleading conclusions, I suggest the statement that the pertaining process leads to reduction of atmospheric carbon dioxide levels (line 483) be removed. (I understand if the authors prefer to not oppose the respective study of LanzaTech et al.).

Comments on the Quality of English Language

English language is overall good. I have suggested some minor edits.

Reviewer 2 Report

Comments and Suggestions for Authors

In this study, the heterologous production of isopropanol was successfully achieved through plasmid-based expression in Acetobacterium woodii. The collaborative action of enzymes leads to a maximum isopropanol production of 5.64 ± 1.08 mM, utilizing CO2 + H2 as carbon and energy sources. This research improved the isopropanol production by employing the genes thlA, ctfAlctfB, adc and sadH. This study is interesting and meaningful. However, there are some issues should be addressed to enhance the quality of the manuscript.

1) Although there is an introduction to the functions and biological characteristics of isopropanol, it is necessary to briefly summarize the existing production methods of isopropanol. This may help emphasize the importance of the proposed alternative method of using acetyl-producing bacteria.

2) Why is it important to use acetyl-producing bacteria to produce isopropanol? Emphasize any potential environmental or economic advantages compared to traditional petroleum methods.

3) Line 215-221 in page 7, please complete the data figures in the section describing the experimental results to present them more clearly.

4) It is suggested that the author moves the initial paragraphs of the discussion section to the introduction. This ensures the inclusion of crucial historical context and challenges in the introduction to better guide readers into the subject.

5) It is recommended that the author improve the discussion section and extract a concise and comprehensive summary, emphasizing the key findings of this study.

6) The authors should address grammar and language issues in this manuscript.

Comments on the Quality of English Language

The authors should address grammar and language issues in this manuscript.
